# Adaptive Temporal-Difference Learning for Policy Evaluation with Per-State Uncertainty Estimates

**Hugo Penedones** *
DeepMind

**Carlos Riquelme** *
Google Brain

**Damien Vincent**
Google Brain

**Hartmut Maennel**
Google Brain

**Timothy Mann**
DeepMind

**André Barreto**
DeepMind

**Sylvain Gelly**
Google Brain

**Gergely Neu**
Universitat Pompeu Fabra

## Abstract

We consider the core reinforcement-learning problem of on-policy value function approximation from a batch of trajectory data, and focus on various issues of Temporal Difference (TD) learning and Monte Carlo (MC) policy evaluation. The two methods are known to achieve complementary bias-variance trade-off properties, with TD tending to achieve lower variance but potentially higher bias. In this paper, we argue that the larger bias of TD can be a result of the amplification of local approximation errors. We address this by proposing an algorithm that adaptively switches between TD and MC in each state, thus mitigating the propagation of errors. Our method is based on learned confidence intervals that detect biases of TD estimates. We demonstrate in a variety of policy evaluation tasks that this simple adaptive algorithm performs competitively with the best approach in hindsight, suggesting that learned confidence intervals are a powerful technique for adapting policy evaluation to use TD or MC returns in a data-driven way.

## 1   Introduction

In reinforcement learning (RL) an agent must learn how to behave while interacting with an environment. This challenging problem is usually formalized as the search for a decision policy—*i.e.*, a mapping from states to actions—that maximizes the amount of reward received in the long run [24]. Clearly, in order to carry out such a search we must be able to assess the quality of a given policy. This process, known as *policy evaluation*, is the focus of the current paper.

A common way to evaluate a policy is to resort to the concept of *value function*. Simply put, the value function of a policy associates with each state the expected sum of rewards, possibly discounted over time, that an agent following the policy from that state onwards would obtain. Thus, in this context the policy evaluation problem comes down to computing a policy's value function.

Perhaps the simplest way to estimate the value of a policy in a given state is to use *Monte Carlo* (MC) returns: the policy is executed multiple times from the state of interest and the resulting outcomes are averaged  [23]. Despite their apparent naivety, MC estimates enjoy some nice properties and have been advocated as an effective solution to the policy evaluation problem [1]. Another way to address the policy evaluation problem is to resort to temporal-difference (TD) learning [22]. TD is based on the insight that the value of a state can be recursively defined based on other states' values [4]. Roughly speaking, this means that, when estimating the value of a state, instead of using an entire trajectory one uses the immediate reward plus the value of the next state. This idea of updating an estimate from another estimate allows the agent to learn online and incrementally.

Both MC and TD have advantages and disadvantages. From a statistical point of view, the estimates provided by MC are unbiased but may have high variance, while TD estimates show the opposite properties [24]. As a consequence, the relative performance of the two methods depends on the amount of data available: while TD tends to give better estimates in small data regimes, MC often performs better with a large amount of data. Since the amount of data that leads to MC outperforming TD varies from problem to problem, it is difficult to make an informed decision on which method to use in advance. It is also unlikely that the best choice will be the same for all states, not only because the number of samples associated with each state may vary but also because the characteristics of the value function itself may change across the state space.

Ideally, we would have a method that adjusts the balance between bias and variance per state based on the progress of the learning process. In this paper we propose an algorithm that accomplishes that by dynamically choosing between TD and MC before each value-function update. *Adaptive TD* is based on a simple idea: if we have confidence intervals associated with states' values, we can decide whether or not to apply TD updates by checking if the resulting targets fall within these intervals. If the targets are outside of the confidence intervals we assume that the bias in the TD update is too high, and just apply an MC update instead.

Although this idea certainly allows for many possible instantiations, in this work we focus on simple design choices. Our experimental results cover a wide range of scenarios, from toy problems to Atari games, and they highlight the *robustness* of the method, whose performance is competitive with the best of both worlds in most cases. We hope this work opens the door to further developments in policy evaluation with function approximation in complex environments.

## 2 The Problem

This section formally introduces the problem of policy evaluation in Markov decision processes, as well as the two most basic approaches for tackling this fundamental problem: Monte-Carlo and Temporal-Difference learning. After the main definitions, we discuss the key advantages and disadvantages of these methods, which will enable us to state the main goals of our work.

### 2.1 Policy Evaluation in Reinforcement Learning

Let $M = \langle \mathcal{S}, \mathcal{A}, P, r, \gamma, \mu_0 \rangle$ denote a Markov decision process (MDP) where $\mathcal{S}$ is the set of states, $\mathcal{A}$ is the set of actions, and $P$ is the transition function so that, for all $s, s' \in \mathcal{S}$ and $a \in \mathcal{A}$, $P(s'|s, a)$ denotes the probability of transitioning to $s'$ from state $s$ after taking action $a$. Also, $r : \mathcal{S} \times \mathcal{A} \to \mathbb{R}$ maps each pair (state, action) to its expected reward, $\gamma \in (0, 1]$ is the discount factor, and $\mu_0$ is the probability distribution over initial states.

Let $\pi : \mathcal{S} \to \mathcal{D}(\mathcal{A})$ be a policy, where $\mathcal{D}(\cdot)$ is the set of distributions over its argument set. Assume at each state $s$ we sample $a \sim \pi(s)$. The value function of $M$ at $s \in \mathcal{S}$ under policy $\pi$ is defined by

$$v^\pi(s) = \mathbf{E} \left[ \sum_{t=0}^{\infty} \gamma^t \, r(S_t, \pi(S_t)) \middle| S_0 = s \right], \qquad (1)$$

where $S_{t+1} \sim P(\cdot|S_t, A_t)$ and $A_t \sim \pi(s)$. We will drop the dependence of $r$ on $a$ for simplicity.

Our goal is to recover value function $v^\pi$ from samples collected by running policy $\pi$ in the MDP. As $\pi$ is fixed, we will simply use the notation $v = v^\pi$ below. We consider trajectories collected on $M$ by applying $\pi$: $\tau = \langle (s_0, a_0, r_0), (s_1, a_1, r_1), \dots \rangle$. Given a collection of $n$ such trajectories $D_n = \{\tau_i\}_{i=1}^n$, a policy evaluation algorithm outputs a function $\widehat{V} : \mathcal{S} \to \mathbb{R}$. We are interested in designing algorithms that minimize the Mean Squared Value Error of $\widehat{V}$ defined as

$$\mathrm{MSVE}(\widehat{V}) = \mathop{\mathbf{E}}_{S_0 \sim \mu_0} \left[ \left( v(S_0) - \widehat{V}(S_0) \right)^2 \right]. \qquad (2)$$

More specifically, we will search for an appropriate value estimate $\widehat{V}$ within a fixed hypothesis set of functions $\mathcal{H} = \{h : \mathcal{S} \to \mathbb{R}\}$, attempting to find an element with error comparable to $\min_{h \in \mathcal{H}} \mathrm{MSVE}(h)$. We mainly consider the set $\mathcal{H}$ of neural networks with a fixed architecture.

The key challenge posed by the policy evaluation problem is that the regression target $v(s)$ in (2) is not directly observable. The algorithms we consider deal with this challenge by computing an

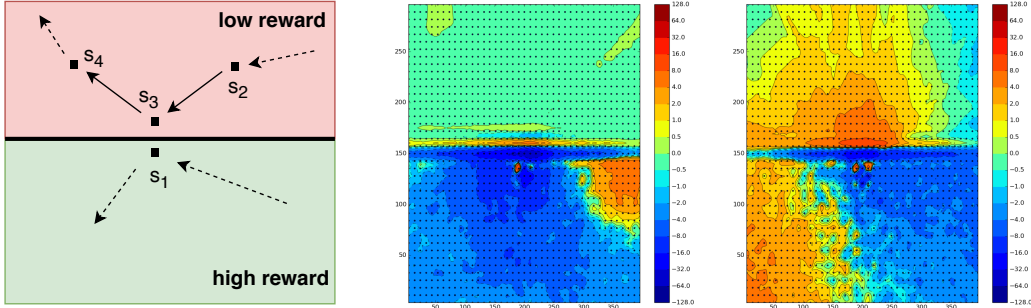

Figure 1: **Left.** Bootstrapping approximation errors. **Center and Right.** MC (center) versus TD (right) on a simple environment with 2 rooms completely separated by a wall (see Map 2 in Figure 11). For each state $s$, the heatmaps show: $\hat{V}(s) - V(s)$. The true value on the upper half of the plane is zero. MC overestimates the values of a narrow region right above the wall, due to function approximation limitations. With TD, these unavoidable approximation errors also occur, but things get worse when bootstrap updates propagate them to much larger regions (see right).

appropriate regression target $T(s)$ and, instead, attempt to minimize $(T(s) - \hat{V}(s))^2$ as a function of $\hat{V}$, usually via stochastic optimization.

## 2.2 Monte Carlo for Policy Evaluation

The Monte-Carlo approach is based on the intuitive observation that the infinite discounted sum of rewards realized by running the policy from a state $s$ is an unbiased estimator of $v(s)$. This suggests that a reasonably good regression target can be constructed for all $s_t \in \tau_i$ as

$$T_{\mathrm{MC}}\left(s_t^{(i)}\right) := \sum_{k=0}^{n_i - t - 1} \gamma^k\, r\left(s_{t+k}^{(i)}\right), \tag{3}$$

where $n_i$ is the length of trajectory $\tau_i$. Thus, one viable approach for policy evaluation is to compute the minimizer within $h \in \mathcal{H}$ of $\sum_{i=1}^{n} \sum_{t=1}^{n_i} (T_{\mathrm{MC}}(s_t^{(i)}) - h(s_t^{(i)}))^2$. Risking some minor inconsistency[2] with the literature, we refer to this method as Monte Carlo policy evaluation.

## 2.3 Temporal-Difference for Policy Evaluation

Temporal-Difference algorithms are based on the fact that the value function should satisfy Bellman equations: $v(s) = r(s) + \gamma \sum_{s'} P(s'|s, a)\, v(s')$ for all $s$, which suggests that a good estimate of the value function should minimize the squared error between the two sides of the above equation. In our framework, this can be formulated as using the regression target

$$T_{\mathrm{TD(0)}}\left(s_t^{(i)}\right) := r\left(s_t^{(i)}\right) + \gamma\, \hat{V}\left(s_{t+1}^{(i)}\right) \tag{4}$$

to replace $v(s)$ in the objective of Equation (2). The practice of using the estimate $\hat{V}$ as part of the target is commonly referred to as "bootstrapping" within the RL literature. Again with a slight abuse of common terminology[3], we will refer to this algorithm as TD(0), or just TD.

TD and Monte Carlo provide different target functions and, depending on the problem instance, each of them may offer some benefits. In the tabular case, it is easy to see that Monte Carlo converges to the optimal solution with an infinite amount of data, since the targets concentrate around their true mean, the value function. However, the Monte Carlo targets can also suffer from large variance due to the excessive randomness of the cumulative rewards. On the other hand, TD can be shown to converge to the true value function in the same setting too [22], with the additional potential to

converge faster due to the potential variance reduction in the updates. Indeed, when considering a fixed value function, the only randomness in the TD target is due to the immediate next state, whereas the MC target is impacted by the randomness of the entire trajectory. Thus, the advantage of TD is more pronounced in low data regimes, or when the return distribution from a state has large variance.

The story may be different with function approximation: even in the limit of infinite data, both MC and TD are going to lead to biased estimates of the value function, due to the approximation error introduced by the function class $\mathcal{H}$. In the case of linear function classes, the biases of the two algorithms are well-understood; the errors in estimating the value functions can be upper-bounded in terms of the distance between the value function and the span of the features, with MC enjoying tighter upper bounds than TD [26, 18]. These results, however, do not provide a full characterization of the errors: even when considering linear function approximation, there are several known examples in the literature where TD provably outperforms MC and vice versa [18]. Thus, the winner between the two algorithms will be generally determined by the particular problem instance we are tackling.

To see the intuitive difference between the behavior of the two algorithms, consider a situation where the true underlying value function $V$ has some sharp discontinuities that the class of functions at hand is not flexible enough to capture. In these cases, both methods suffer to fit the value function in some regions of the state space, even when we have lots of data. The errors, however, behave differently for the two methods: while the errors of Monte Carlo are localized to the regions with discontinuities due to directly fitting the data, TD bootstraps values from this problematic region, and thus propagates the errors even further. We refer to such errors arising due to discontinuities as *leakage*.

To illustrate the leakage phenomenon, consider the toy example in the left side of Figure 2.3. There is a wall separating the green region (high reward), and the red region (low reward). Assume that a function approximator $f$ will need to make a compromise to fit both $s_1$ and $s_3$ as they are close in the state space, even though no trajectory goes through both of them due to the wall. For example, we can assume $f$ will over-estimate the true value of $s_3$. Let us now examine a third state, $s_2$, that is in the red region, and such that there is a trajectory that goes from $s_2$ to $s_3$. The distance in state-space between $s_2$ and $s_3$ could be, in principle, arbitrarily large. If we apply Monte Carlo updates, then the impact of the $s_1 \rightarrow s_3$ leakage on $s_2$ will be minimal. Instead, the TD update for the value of $s_2$ will explicitly depend on the estimate for $s_3$, which is overestimated due to function approximation near a wall. In the center and right side of Figure 2.3, we show the actual outcome of running TD and MC in such a setting. TD dramatically propagates estimation errors far into the low-reward region as expected, while MC is way more robust and errors stay located very close to the wall. Alternatively, in a smoother scenario, however, bootstrapping function approximation estimates can still be certainly beneficial and speed up learning. We present and analyze a toy MDP in the appendix, Section A.

The key observation of our work is that different *regions of the state space* may be best suited for either TD or MC updates, amending the existing folk wisdom that TD and MC may "globally" outperform each other in different MDPs. Accordingly, we set our goal as designing a method that adaptively chooses a target depending on the properties of the value function around the specific state.

## 3  The Adaptive-TD Algorithm

In the previous sections, we saw that whether MC or TD is a better choice heavily depends on the specific scenario we are tackling, and on the family of functions we choose to fit the value function. While in hindsight we may be able to declare a winner, in practice the algorithmic decision needs to be made at the beginning of the process. Also, running both and picking the best-performing one in training time can be challenging, as the compound variance of the return distribution over long trajectories may require extremely large validation sets to test the methods and choose, while this then limits the amount of available training data. We aim to design a *robust* algorithm which does not require any knowledge of the environment, and that dynamically adapts to both the geometry of the true value and transition functions, and to the function approximator it has at its disposal.

In this section, we propose an algorithmic approach that aims to achieve the best of both the MC and TD worlds. The core idea is to limit the bootstrapping power of TD to respect some hard limits imposed by an *MC confidence interval*. The main driver of the algorithm is TD, since it can significantly speed up learning in the absence of obstacles. However, in regions of the state space where we somehow suspect that our estimates may be poor (e.g., near walls, close to big rewards, non-markovianity, partial observability, or after irrecoverable actions) we would like to be more

**Input:** Confidence level $\alpha \in (0, 1)$. Trajectories $\tau_1, \ldots, \tau_n$ generated by policy $\pi$.

---

Let $S$ be the set of visited states in $\tau_1, \ldots, \tau_n$. Initialize $\widehat{V}(s) = 0$, for all $s$.
Compute Monte-Carlo returns dataset as in (3): $D_{\text{MC}} = \{(s, T_{MC}(s))_{s \in S}\}$.
Fit confidence function to $D_{\text{MC}}$: $\text{CI}_{MC}^\alpha(s) := (L_{MC}^\alpha(s), U_{MC}^\alpha(s))$.
**repeat**
   **for** $i = 1$ **to** $n$ **do**
      **for** $t = 1$ **to** $|\tau_i| - 1$ **do**
         $s_t^{(i)}$ is the $t$-th state of $\tau_i$.
         $T_{\text{TD}(0)} = r(s_t^{(i)}) + \gamma \, \widehat{V}(s_{t+1}^{(i)})$
         **if** $T_{\text{TD}(0)} \in (L_{MC}^\alpha(s_t^{(i)}), U_{MC}^\alpha(s_t^{(i)}))$ **then**
            $T_{i,t} \leftarrow T_{\text{TD}(0)}$
         **else**
            $T_{i,t} \leftarrow (L_{MC}(s_t^{(i)}) + U_{MC}(s_t^{(i)})/2$
         **end if**
         Use target $T_{i,t}$ to fit $\widehat{V}(s_t^{(i)})$.
      **end for**
   **end for**
**until** epochs exceeded

---

**Algorithm 1:** Adaptive TD

conservative, and rely mostly on targets purely based on data. We explicitly control how conservative we would like to be by tuning the *confidence level* $\alpha$ of the intervals: by letting $\alpha$ go to 1, the intervals become trivially wide, and we recover TD. Similarly, if we let $\alpha$ go to 0, we end up with MC.

It is easy to design toy environments where none of the obstacles described above apply, and where TD is the optimal choice (see Map 0 in Figure 11). As a consequence of the switching and testing cost, it is not reasonable to expect our algorithm to dominate *both* MC and TD in *all* environments. Our goal is to design an algorithm that is not worse than the worst of MC and TD in any scenario, and is close to the best one in most cases, or actually better. When data goes to infinity and the true value function falls in our family of models, the MC intervals will converge to the true values and Adaptive TD will be forced to converge too. However, these are asymptotic results.

## 3.1 Formal Description

Adaptive TD is presented as Algorithm 1. It has two components: confidence computation, and value function fitting. First, we compute the MC target dataset $D_{\text{MC}} = \{(s, T_{MC}(s))_{s \in S}\}$ for all states $s$ in any input episode (we refer to $S$ as the union of those). Then, we need to solve the regression problem with a method that provides confidence intervals: with probability $\alpha$ the expected return from $s$ under $\pi$ is in $(L_{MC}^\alpha(s), U_{MC}^\alpha(s))$. There are a variety of approximate methods we can use to compute such confidence bounds; we discuss this in detail in the next subsection. Note, however, that this can be regarded as an additional input or hyper-parameter to Adaptive TD.

In the second stage, after fixing a confidence function $\text{CI}_{MC}^\alpha(s)$, we apply a constrained version of TD. We loop over all states $s \in S$ (possibly, in a randomized way, as the main loop over data can be replaced by a stochastic mini-batch), and we compute their TD target $T_{\text{TD}(0)}(s)$. If the TD target falls within the confidence interval $\text{CI}_{MC}^\alpha(s)$, we simply use it to update $\widehat{V}(s)$. If it does not, i.e. when $T_{\text{TD}(0)}(s) \notin (L_{MC}^\alpha(s), U_{MC}^\alpha(s))$, then we no longer trust the TD target, and use the mid-point of the MC interval, $(L + U)/2$. We can also use the closest endpoint to $T_{\text{TD}(0)}(s)$, either $L$ or $U$.

## 3.2 Uncertainty Estimates

Poor quality uncertainty estimates may *severely* affect the performance of Adaptive TD. In particular, in those states where the ground truth is not captured by the MC confidence intervals, the TD target will be *forced* to bootstrap wrong values without any possibility of recovery. In the case of neural networks, uncertainty estimation has been an active area of research in the last few years. In some

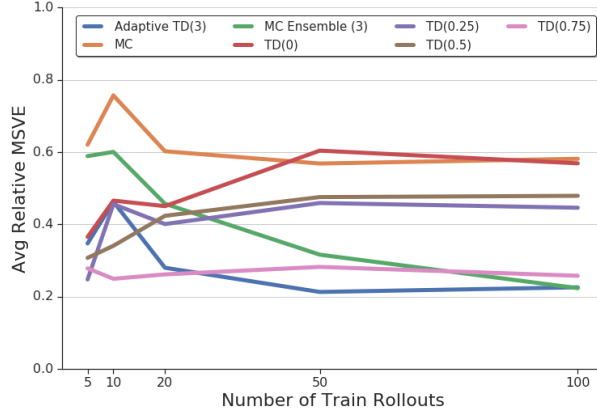

Figure 2: Average normalized MSVE for Lab2D and Atari environments in each data regime. For each number of train rollouts and scenario, we normalize the MSVE of each algorithm $A$ by $(\text{MSVE}(A) - \min_{A'} \text{MSVE}(A'))/(\max_{A'} \text{MSVE}(A') - \min_{A'} \text{MSVE}(A'))$. Equivalently, the worst algorithm is assigned relative MSVE 1, and the best one is assigned relative MSVE 0. Then, for each number of rollouts, we take the average across scenarios (i.e., all the 10 environments are worth the same). This allows for a reasonably fair comparison of performance in different domains.

decision-making tasks, like exploration, all we need are *samples* from the output distribution, while actual intervals are required for Adaptive TD. A simple fix for models that provide samples is to take a number of them, and then construct an approximate interval.

Common approaches include variational inference [5], dropout [9], Monte Carlo methods [27, 12], bootstrapped estimates [8, 15], Gaussian Processes [16], or Bayesian linear regression on learned features [19, 17, 2]. While all the methods above could be used in combination with Adaptive TD, for simplicity, we decided to use an ensemble of $m$ MC networks [11]. The algorithm works as follows. We fit $m$ networks on the $D_{\text{MC}} = \{(s, T_{MC}(s))_{s \in S}\}$ dataset (we may or may not boostrap the data at the episode level). Given a new state $s$, the networks provide value estimates $v_1, \ldots, v_m$ at $s$. We then compute a *predictive* confidence interval, under the assumption that $v_i$ for $i = 1, \ldots, m$ are i.i.d. samples from some distribution $\mathbf{F}$. Now, if $v_{m+1}$ was sampled from the same distribution, then we could expect $v_{m+1}$ to fall in the predictive interval with probability $\alpha$. The idea is that the TD estimate should approximately correspond to another sample from the MC distribution. If the deviation is too large, we will rely on the MC estimates instead.

In particular, we do assume $\mathbf{F}$ is Gaussian: $v_1, \ldots, v_m \sim \mathcal{N}(\mu, \sigma^2)$ for unknown $\mu, \sigma^2$. Let us define $\bar{v} = \sum_i v_i/m$, and $\hat{\sigma}_m^2 = \sum_i (v_i - \bar{v})^2/(m-1)$. Finally, if the assumptions hold, we expect that

$$\bar{v} - z_\alpha \hat{\sigma}_m \sqrt{1 + 1/m} \leq v_{m+1} \leq \bar{v} + z_\alpha \hat{\sigma}_m \sqrt{1 + 1/m} \tag{5}$$

with probability $\alpha$, where $z_\alpha$ is the $100(1 - \alpha/2)$ percentile of the Student's distribution with $m - 1$ degrees of freedom. Then, we set $L_{MC}^\alpha(s)$ and $U_{MC}^\alpha(s)$ to the left and right-hand sides of (5) (note $v_i$ depends on $s$). Of course, in practice the assumptions may not hold (for example, $v_i, v_j$ for $i \neq j$ will not be independent unless we condition on the data), but we still hope to get a reasonable estimate.

## 4 Experimental Results

In this section we test the performance of Adaptive TD in a number of scenarios that we describe below. The scenarios (for which we fix a specific policy) capture a diverse set of aspects that are relevant to policy evaluation: low and high-dimensional state spaces, sharp value jumps or smoother epsilon-greedy behaviors, near-optimal and uniformly random policies. We present here the results for Labyrinth-2D and Atari environments, and Mountain Car is presented in the appendix, Section C.

We compare Adaptive TD with a few baselines: a single MC network, raw TD, and TD($\lambda$). TD($\lambda$) is a temporal differences algorithm which computes an average of all $n$-step TD returns (an extension of the 1-step target in (4)), [23]. For a clean comparison across algorithms in each scenario, we normalize the MSVE of all algorithms ($y$-axis) by the worst performing one, and we do this independently for

each number of data rollouts ($x$-axis). In addition, the appendix contains the absolute values with empirical confidence intervals for all cases. Our implementation details are presented in Section B of the appendix. In general, we did not make any effort to optimize hyper-parameters, as the goal was to come up with an algorithm that is robust and easy to tune across different scenarios. Accordingly, for Adaptive TD, we use an ensemble of 3 networks trained with the MC target, and confidence intervals at the 95% level. The data for each network in the ensemble is bootstrapped at the rollout level (i.e., we randomly pick rollouts with replacement). Plots also show the performance of the MC ensemble with 3 networks, to illustrate the benefits of Adaptive TD compared to its auxiliary networks.

**Labyrinth-2D.** We first evaluate the performance of the algorithms in a toy scenario which represents a 2-d map with some target regions we would like to reach. The state $s = (x, y)$ are the coordinates in the map, and the policy takes a uniformly random angle and then applies a fixed-size step. The initial state $s_0$ for each episode is selected uniformly at random inside the map, and the episode ends after each step with probability $p = 0.0005$. Reward is $r = 30$ inside the green regions, $r = 0$ elsewhere. The maps layouts and their value functions are shown in Figure 11 in the appendix. The simple different layouts cover a number of challenging features for policy evaluation: sharp jumps in value near targets, several kind of walls, and locked areas with no reward (see maps 2 and 3). Due to the randomized policy and initial state, we tend to uniformly cover the state space. We run experiments with $n = 5, 10, 20, 50, 75, 100$ training episodes. We approximate the ground truth in a grid by sampling and averaging a large number of test episodes from each state.

The results are shown in Figure 3. As expected, in most of the maps we observe that MC outperforms TD in high-data regimes, while MC consistently suffers when the number of available data rollouts is limited. Adaptive TD shows a remarkably robust performance in all cases, being able to strongly benefit from TD steps in the low-data regime (see, basically, all maps) while remaining very close to MC's performance when a large number of rollouts are available. In that regime, the improvement with respect to TD is dramatic in maps that are prone to strong leakage effects, like maps 1, 2, 3, and 5. In Figure 14 in the appendix, we can also see the results for TD($\lambda$). In this particular case, it seems $\lambda = 0.75$ is a good choice, and it is competitive with Adaptive TD in challenging maps 1, 2, 3, and 5. However, the remaining values of $\lambda$ are mostly outperformed in these scenarios. Figure 17 shows the regions of the maps state space where the TD target falls outside the MC interval for Adaptive TD.

**Atari.** The previous examples illustrate many of the practical issues that arise in policy evaluation. In order to model those issues in a clean disentangled way, and provide some intuition, we focused so far on lower-dimensional state spaces. In this section we evaluate all the methods in a few Atari environments [3]: namely, Breakout, Space Invaders, Pong, and MsPacman. The state consists of four stacked frames, each with $(84, 84)$ pixels, and the initial one is fixed for each game. We would like to focus on competitive policies for the games, while still offering some stochasticity to create a diverse set of trajectories (as Atari environments are deterministic). We use soft Q policies that sample from the action distribution to generate the training and test trajectories. The temperature of the softmax layer was adjusted to keep a good trade-off between trajectory diversity and performance of the policy. Directly computing the ground-truth value is not feasible this time, so we rely on a large number of test rollouts to evaluate our predictions. This increases the variance of our MSVE results.

The results are shown in Figure 4. TD does a good job for all number of rollouts. This suggests that in high-dimensional state spaces (like Atari frames) the required number of samples for MC to dominate may be extremely large. In addition, a single MC network seems to struggle in all games, while the prediction of the MC ensemble proves significantly more robust. Adaptive TD outperforms MC, and its auxiliary MC ensemble. Moreover, it offers a performance close to that of TD, maybe due to wide MC confidence intervals in high-dimensional states which reduce Adaptive TD to simply TD in most of the states. We show the results for TD($\lambda$) in Figure 18 in the appendix. In this case, $\lambda = 0.75$ –which did a good job in Labyrinth-2D scenarios– is always the worst. In particular, Adaptive TD improvements compared to TD($\lambda = 0.75$) range from 30% in the low-data regimes of Pong, to consistent 20% improvements across all data regimes of Space Invaders.

**Summary.** Figure 2 displays the overall results normalized and averaged over the 10 scenarios. Adaptive TD strongly outperforms TD and MC, and offers significant benefits with respect to its auxiliary ensemble, and TD($\lambda$). This highlights the main feature of Adaptive TD: its *robustness*. While TD and MC outperform each other often by a huge margin depending on the scenario and data size, Adaptive TD tends to automatically mimic the behavior of the best-performing one. TD($\lambda$) methods offer a way to interpolate between TD and MC, but they require to know a good value of $\lambda$

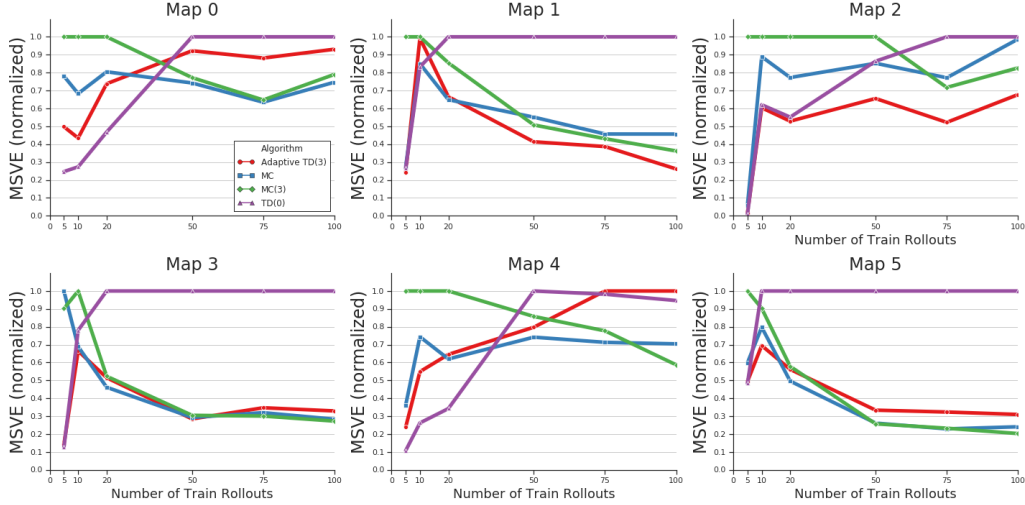

Figure 3: **Labyrinth-2D.** For each number of train rollouts, we normalize the MSVE of each algorithm $A$ by $\mathrm{MSVE}(A)/\max_{A'} \mathrm{MSVE}(A')$. Absolute numbers and conf. intervals in the appendix.

in advance, and we have seen that this value can significantly change across problems. In most cases, Adaptive TD was able to perform –at least– comparably to TD($\lambda$) for the *best* problem-dependent $\lambda$.

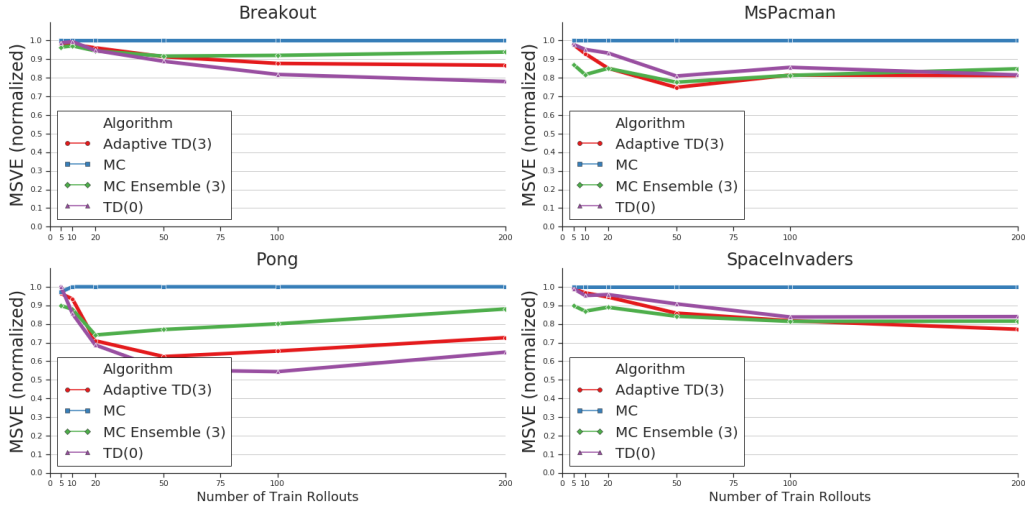

Figure 4: **Atari.** For each number of train rollouts, we normalize the MSVE of each algorithm $A$ by $\mathrm{MSVE}(A)/\max_{A'} \mathrm{MSVE}(A')$. Absolute numbers and confidence intervals are in the appendix.

## 5   Related Work

Both $n$-step TD and TD($\lambda$) offer practical mechanisms to control the balance between bias and variance by tuning $n$ and $\lambda$. However, these parameters are usually set *a priori*, without taking into account the progress of the learning process, and are not state-dependant.

A number of works have addressed on-policy evaluation. In [13] the authors introduce an algorithm for batch on-policy evaluation, which is capable of selecting the best $\lambda$ parameter for LSTD by doing efficient cross-validation. However, it only works for the case of linear function approximation, and does not take per-state decisions. In the TD-BMA algorithm [7], decisions are taken by state, like in ours, while TD-BMA is restricted to the tabular setting. We cover function approximation, including deep neural networks, and large dimensional input spaces, where uncertainty estimates require different techniques.

Per-state $\lambda$ was used for the off-policy case in [20]. (For an overview of methods with fixed $\lambda$, including the off-policy case, see [10].) This study is continued in [21] with methods that selectively (de-)emphasize states. The motivation is that function approximation has to give a compromise function (e.g. forcing similar values at nearby states, even if the observed values are not similar), which should be guided by emphasizing the more important states. In off-policy evaluation (the focus of their paper) there is more interest in states that will occur more frequently in our actual policy. Similarly, when we switch to MC for some states in our approach, we may be less interested to model their value correctly in the TD function approximation. This paper also hints that $\lambda(s)$ may be modified depending on the variance of the returns after $s$; this is then developed in [28].

The algorithm of [28] estimates the variance of the $\lambda$–returns arising from the data by establishing a Bellman operator for the squared return, for which they are looking for a fixed point. Then they optimize "greedily" the $\lambda(s_t)$ such that they get the optimal bias/variance trade-off for this state. However, their variance of the returns is restricted to the uncertainty coming from the actions and returns, but does not take into account the model uncertainty arising from the function approximation (which we include here by evaluating an ensemble of networks). [25] introduced a different approach to compute an optimized and state-dependent combination of $n$–step–returns. They ask what the optimal combination of $n$–step returns would be if the estimates were unbiased and their variances and covariances were known. This differs from our approach as we are also trying to minimize the bias that is introduced by function approximation and that is amplified by TD's bootstrapping.

## 6 Future Work

There are a number of avenues for future work. Adaptive TD can be easily extended to $n$-step TD or TD($\lambda$): at each transition, the agent checks whether the potential $n$-step or TD($\lambda$) target is within the associated confidence interval. If so, the agent applies the TD update; otherwise, it proceeds to the next transition and carries over the unused target to compute the next TD target and repeat the process. In addition, the proposed policy evaluation algorithm can be implicitly or explicitly incorporated into a policy improvement one for control. Finally, we expect that constructing more sophisticated and accurate confidence intervals based on MC returns will improve the performance of Adaptive TD.

**Acknowledgments**

The authors thank Matthieu Geist for his comments and suggestions, and the anonymous reviewers for their valuable feedback. G. Neu was supported by "La Caixa" Banking Foundation through the Junior Leader Postdoctoral Fellowship Programme and a Google Faculty Research Award.

## Footnotes

*These two authors contributed equally. Correspondence to rikel@google.com

[2][23, 24] exclusively refer to the tabular version of the above method as Monte Carlo; this method is a natural generalization to general value-function classes.

[3]This algorithm would be more appropriately called "least squares TD" or LSTD, following [6], with the understanding that our method considers general (rather than linear) value-function classes.

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
