[Supplementary Material · Adaptive_Temporal_Differences_Learning___Neurips_long.pdf]

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

# A  A simple example

In this section we illustrate the different bias-variance tradeoffs achieved by Monte Carlo and TD through a simple example, particularly highlighting the leakage propagation effect of TD described less formally in the previous section.

Consider the following MDP, with one initial state $s_0$ where $k > 1$ actions are available. Each action $a_i$ results in a deterministic transition to the corresponding state $s_i$. The first $p$ of these states $s_1, \ldots, s_p$ then transfer the agent deterministically to state $b_1$, and the remaining states $s_{p+1}, \ldots, s_k$ are deterministically followed by $b_2$. States $b_1$ and $b_2$ are followed by another deterministic transition to state $q$ and then to a final state, emitting a random $\mathcal{N}(\mu, \sigma^2)$ reward. All other transitions lead to no reward. The episode ends when the final state is reached.

Figure 5: Simple episodic Markov Decision Process.

Let $\pi$ be the uniform policy that, at $s_0$, chooses actions $a_{1:k}$ leading to $s_{1:k}$ with equal probability. We assume the discount factor is $\gamma = 1$. The true value function is $v_\pi(s) = \mathbf{E}[R] = \mu$ for all $s \in \mathcal{S}$. Suppose we need to estimate the value function from a set of $n$ episodes collected from $\pi$ with final rewards $r_1, \ldots, r_n$.

Let us compare the performance of both the MC update (3) and the TD(0) target (4) when no function approximation is used, that is, when $\widehat{V}$ can take any real value for any state. Note all $n$ trajectories pass through state $q$, but on average only $n/k$ pass through each of the intermediate states $s_1, ..., s_k$. The estimate for $q$ will be equal for MC and TD: $\widehat{V}(q) = \sum_i r_i/n$. Its distribution has mean $\mathbf{E}[\widehat{V}(q)] = \mathbf{E}[R] = \mu$ and variance $\mathrm{Var}[\widehat{V}(q)] = \sigma^2/n$.

On the other hand, the variance of $\widehat{V}$ for states $s_{1:k}$ does differ significantly for TD and MC. The variance of the MC estimate has variance $\mathrm{Var}[\widehat{V}_{\mathrm{MC}}(s_i)] \approx k\sigma^2/n$, due to approximately[4] $n/k$ episodes going through each $s_{1:k}$. However, the TD estimator correctly realizes that $\widehat{V}(s_i) = \widehat{V}(q)$, thus inheriting only the variance of the estimator in state $q$: $\mathrm{Var}[\widehat{V}_{\mathrm{MC}}(s_i)] = \sigma^2/n$. Thus, TD reduces the variance of the estimates by a factor of $k$.

In the above setting with no function approximation, both methods are unbiased, and TD will dominate over MC for any finite size amount of data due to its reduced variance, while both converge to the same solution in the limit of infinite data. However, assume now we use a function approximator that is *not* able to correctly represent the value $\mu$ for either state $b_1$ or $b_2$, as it could happen in a general scenario with overly smooth function approximators. TD estimates will introduce and propagate the bias to some of the states $s_1, ..., s_k$, while MC will not (noting though that for more general

Figure 6: Mean-squared Value Error (MSVE) for MC, TD, and Adaptive TD as a function of the data size. The left plot shows the tabular setting; the right plot shows the function approximation case with a fixed bias. Each point is run 20 times to smooth out estimates, and $\mu = 0$.

Figure 7: MC and TD estimates for the example of Section A ($\mu = 0$) with a biased function approximator. Adaptive TD detects and corrects the mismatch between the TD updates and the MC confidence intervals.

function approximators, MC estimates in some states may also influence estimates in others). In this case, depending on the relative magnitude of TD's bias and MC's variance, and the amount of available data, one or the other will offer better performance. Figure 6 shows the performance of the two algorithms in the two settings, highlighting that the bias encoded in the function approximator impacts TD more severely than MC due to leakage propagation, when the number of sample episodes is high enough. We report the MSVE for the intermediate $s_{1:k}$ states. Notably, despite leakage, TD still outperforms MC when less data is available, due to its reduced variance.

### A.1 Results

We clearly see in Figure 6 that Adaptive TD is able to take advantage from TD when the MC variance is too large (left), while avoiding bootstrapping erroneous values by detecting and correcting suspicious TD-estimates in the function approximation case (right). Moreover, in Figure 7, we see how in the latter case TD updates mostly fall outside the 95% Monte Carlo confidence intervals.

# B  Implementation Details and Neural Network Hyper-Parameters

Our experiments in the Labyrinth-2D and Mountain Car environments, which both have 2d state spaces, were conducted using a multi-layer perceptron with exactly the same configuration. For Atari, we followed the standard pre-processing from the DQN paper [14], where the inputs are reduced to 84x84, and 4 consecutive frames are stacked. The architecture of the convnet used is also standard, with the exception that instead of 18 outputs, we only have 1, as we are estimating state-value functions, not action-value functions. Details are provided in Table 1 below.

The implementation of TD does *not* use a target network; we consider the current estimates for the target (while we do not optimize with respect to the target, i.e., we apply a stop-gradient operation). In general, we did not make any effort to optimize hyper-parameters.

Table 1 summarizes the parameters of the neural networks used as value function approximators and the training hyper-parameters used in the experiments on the 2D environments and ATARI. The parameters for ATARI are the same as the ones from the original DQN paper [14].

| **Neural network** | | |
|---|---|---|
| | 2D Envs. | Atari |
| Input dimensions | 2 | 84 x 84 x 4 |
| Convnet output channels | - | (32, 64, 64) |
| Convnet kernel shapes | - | (8, 4, 3) |
| Convnet strides | - | (4, 2, 1) |
| Linear layers hidden-units | (50, 50) | (512) |
| Non-linearities | Relu | Relu |
| **Training hyper-parameters** | | |
| Mini-batch size | 512 | 32 |
| Optimiser algorithm | Adam | Adam |
| Learning rate | 0.001 | 0.0000625 |
| Beta1 | 0.9 | 0.9 |
| Beta2 | 0.999 | 0.999 |
| Epsilon | 1e-08 | 0.00015 |
| Training batches | 50000 | 250000 |

Table 1: Neural network architectures and hyper-parameter details.

## B.1  Online Scenarios

For policy evaluation, we assume all data is collected in advance. Sometimes this assumption may be too strong, and we would also like to allow for updates in our confidence intervals based on a stream of new data. When the data comes from the same policy, the extension should be straightforward. In general, we can train both estimates in parallel (say, an MC ensemble, and a TD network), and freeze both of them every fixed number of updates as it is nowadays standard in DQN (target network) to keep training a copy of the TD network. We leave the exploration of these extensions as future work.

# C   Mountain Car Environment

We also test the algorithms in the popular Mountain Car environment, where the goal is to control a car in order to climb a steep hill. The state has two coordinates, corresponding to the velocity and position of the car, and there are three actions: move left, move right, and do nothing. We use a near-optimal policy together with $\epsilon$-greedy steps, for $\epsilon = 0.2$. We compute the ground truth value function as in the previous case, see Figure 8 below.

Figure 8: Mountain Car value function for near-optimal policy with epsilon greedy ($\epsilon = 0.2$) actions.

The results are shown in Figure 9 (left). In this case MC seems to consistently outperform TD. Adaptive TD offers strong performance in the low-data regime, and it mimics the behavior of MC when more data is available (and, presumably, we have access to better confidence intervals). The gains with respect to TD are significant. Figure 9 (right) shows TD($\lambda$) methods with $\lambda \geq 0.5$ are competitive with the MC ensemble and with Adaptive TD.

Figure 9: **Mountain Car.** For each number of train rollouts, we normalize the MSVE of each algorithm $A$ by $\mathrm{MSVE}(A)/\max_{A'} \mathrm{MSVE}(A')$.

Finally, for completeness, we show in Figure 10 the unnormalized version of the plots in Figure 9.

Figure 10: MSVE for the Mountain Car environment. Confidence intervals over 20 seeds.

# D Labyrinth-2D Environments

We created a set of six 2D maps used as toy environments in this paper. The layout of those environments always includes at least one reward represented as a green disk on Figure 11. The ground truth value function of a random policy navigating in those environments is shown on the same Figure. This ground truth is used as a reference to compute the MSVE of the adaptive TD algorithm as well as the other baselines (Figure 14). In Figure 17, we also present some insights on the TD versus MC decisions of the adaptive TD algorithm. We can clearly see that TD is selected for most of the states but the ones next to the wall where MC is preferred to prevent further leaking of the approximation errors observed near the wall.

Figure 11: True value functions for the uniformly random policy in Labyrinth-2D: Maps 0, 1 and 2 (Left), Maps 3, 4, 5 (Right).

In addition, we solved the same set of scenarios with a different function approximator: a simple piece-wise constant function on a 2D grid, where all the states within a cell are assigned the same value. Every map layout has size 400 x 300 and the grid layout chosen was of cells of size 19 x 19. This ensures that there was no easy coincidence with the grid layout and the position of the walls.

Figure 12: **Labyrinth-2D.** TD($\lambda$) results. For each number of train rollouts, we normalize the MSVE of each algorithm $A$ by $\mathrm{MSVE}(A)/\max_{A'}\mathrm{MSVE}(A')$.

Figure 13: **Labyrinth-2D.** Unnormalized MSVE results. Confidence intervals over 20 seeds.

Figure 14: **Labyrinth-2D.** Unnormalized MSVE results for TD($\lambda$). Confidence intervals over 20 seeds.

Figure 15: **Labyrinth-2D.** Algorithms use piece-wise constant approximation functions on a 2D grid. For each number of train rollouts, we normalize the MSVE of each algorithm $A$ by $\mathrm{MSVE}(A) / \max_{A'} \mathrm{MSVE}(A')$.

Figure 16: Visualization of the quality of the confidence intervals learned with the neural network ensembles. We take the ground truth value function and check for every state whether it falls inside the confidence interval (green color) or otherwise (red for over-estimation and blue for under-estimation). This shows that overall, the confidence intervals are reasonable.

Figure 17: Adaptive TD confidence interval violations per state in the Labyrinth 2D map layouts, after 300 iterations of training, in the regime with 50 training rollouts. Green means the TD estimate is inside the MC confidence interval for that state. Red regions are over-estimations and blue regions under-estimations. We see that corrections are needed in significant regions of the space at this stage.

# E   Atari Environments

For each of the 4 Atari games, we took a policy that was trained using DQN [14] on that game, and generated data by running it and sampling actions according to the softmax distribution of the output layer, with temperature = 0.02. Each episode was limited to a maximum of 4000 steps. Training was done using varying amounts of episodes, from 5 to 200, as illustrated by the plots. For evaluation, in all cases, we used a disjoint set of 100 episodes generated using the same procedure. We use a discount factor of 0.99 in all experiments.

Figure 18:   **Atari.** TD($\lambda$) results. For each number of train rollouts, we normalize the MSVE of each algorithm $A$ by $\mathrm{MSVE}(A)/\max_{A'} \mathrm{MSVE}(A')$.

Figure 19: **Atari.** Unnormalized MSVE results for main baselines. Confidence intervals over 20 seeds.

Figure 20: **Atari.** Unnormalized MSVE results for TD($\lambda$). Confidence intervals over 20 seeds.