[Reviews · NeurIPS 2019]

Reviewer 1



i. Novelty: While the idea of combining TD and MC methods is novel, the entire paper mostly provides only empirical results but falls short in terms of the theoretical contribution which seems to be not very well suited for this particular venue. ii. Writing: The paper is overall well written although I would suggest to state the proposed algorithm (Alg-1) in the main paper, moving description of the MC (Sec 2.2) and TD (Sec 2.3) policy evaluation method to the appendix.

Reviewer 2



************ [updated after the rebuttal] I am quite positive about this paper, robust approaches are important in practice. I understand that for heuristic approaches like this, proving theoretical guarantees is hard. The authors at least give some intuition about its asymptotics in the rebuttal, which should be easily formalized (and I would request it to be added to the paper if accepted). The authors did a very good job argumenting for their approach and evaluating its performance numerically, on realistic examples. However, NeurIPS might not be the best venue for such work. It only covers evaluation of a policy. Perhaps a follow-up paper which covers incorporating this approach into optimization (policy improvement), which the authors say is left for future research, would be more suitable. If the paper is rejected from NeurIPS, I would like to encourage the authors to add some (even if weak or asymptotical) theoretical guarantees and submit it elsewhere. Also some of the approaches mentioned in the books by Warren Powell (on Approximate Dynamic Programming and on Optimal Learning) might help the authors to refine or support their approach, or serve as additional benchmarks. ************ Originality: I can't judge since I am not familiar with work in this area. Quality: The proposed algorithm is designed based on in-depth understanding of TD and MC approaches and on their strengths and weaknesses. Clarity: very good, easy to read, no typos. Significance: A potentionally useful algorithm as shown on a wide range of problem instances, but its significance hasn't been evaluated by benchmarking with other methods. I have not identified ay minor comments such as unclear sentences or typos.

Reviewer 3



Summary. The authors propose a novel method for adaptively using either the MC method for policy evaluation or the temporal difference method. The authors aim to solve the problem of balancing bias and variance in the reinforcement learning setting and to this end propose the Adaptive TD algorithm. The algorithm takes as input a set of sample episodes which it uses to bootstrap some confidence intervals for the value function of each state. It then compares the TD estimate for each of these states with these confidence intervals and keeps the TD estimate if it fits inside, otherwise, it picks the middle of the confidence interval as it assumes the TD estimate is essentially biased and inaccurate. The process repeats for a number of epochs (since the TD estimates change as the value function estimate for the future state is updated by the adaptive-TD rule). I think this paper shows promise: the method is, to my knowledge, original and from the numerical experiments seems to achieve the target the authors set for it - dominating TD and MC in the worst case. I think there is room to build on top of this method as well. However, overall, I found the presentation of the paper to be insufficiently clear, with key points not being sufficiently addressed (see below) and the method to have limited practical applicability. I therefore vote to reject this paper in its current form. Detailed comments: It is hard to get excited about this paper. I like the idea of using confidence intervals on the MC estimate to figure out whether or not the TD estimate is reliable. To my knowledge, this is original and makes a lot of sense. One major concern I have with this paper is the fact that the computational complexity increases with time. I think this severely limits the practical relevance of the method. The presentation, in my opinion is poor and the method and its relevance are very hard to understand from the paper. I found the choice of putting the pseudocode of the algorithm in the Appendix to be particularly uninspired. I think the setting is not presented sufficiently clearly: what is the practical problem that is solved here? Is this new way of estimating value functions supposed to be used after every time step n of a RL algorithm or is it a way of approximating the optimal policy after training is complete? It seems that running this algorithm throughout the execution of a RL agent is prohibitively expensive (several epochs over all observations so far). I have had a hard time finding the answers to the following questions: 1) The confidence intervals are computed based on inference of m neural networks trained on the MC target. In my opinion, the difficulties of this training step are not discussed sufficiently. If the MC target is not reliable enough to be used why would these networks' inferred estimates be more reliable? Another insufficiently discussed aspect is the fact that the confidence intervals assume these estimates are i.i.d. gaussians (which is a fairly bold assumption in this case). How much is the algorithm affected by this assumption - can we assume whatever distribution here without degrading numerical performance? Using a GP (as the authors indicate would be remedy) or other such techniques might offer the comfort of reliable confidence intervals, but further increase the already large (in my opinion) computation cost. Putting a confidence interval on the output of a neural network trained at runtime is far from a trivial task, which brings us to point 2). 2) In the experiments, m = 3. This would lead to very large confidence intervals. How are m, \alpha and the confidence threshold chosen? If for example we pick m to be large, this would create very small confidence intervals which would reduce the algorithm to always using MC estimates (since the TD estimate is always outside)? This seems like an arbitrary way of getting confidence intervals (these intervals should depend on the number of sampled states n, not m - handpicked in the algorithm design phase) that heavily relies on the parameter choice. 3) What are the drawbacks of this approach? I do not think the authors address these sufficiently. In the experiments on Lab2D - Figure 2, it seems MC ensemble(3) is not done learning. Would it outperform Adaptive TD(3) given more time? What is the relative runtime between the algorithms? In Figure 3, Adaptive TD(3) behaves very poorly on 2 of the 6 maps. This shows that its performance is not reliable and highly problem dependent - can these problems be identified in advance? Figure 4 does however show better performance. The figures in the appendix also show the algorithm achieves the goal set by the authors: dominate both TD and MC in the worst case over a large number of settings. I find Figure 1 to be uninformative and could be sacrificed to include the algorithm pseudocode in the main body of the paper. ================ Post-Rebuttal ================ I am happy with the clarifications made by the authors. I now have a better understanding regarding what exactly are the effects of increasing $m$ on the confidence intervals on the MC estimate. I still find it odd that numbers this small (like m=3) provide usable confidence intervals. Regarding the "computational complexity increasing with time", I was assuming the algorithm is used in a reinforcement learning loop. In consequence, I updated my score to 6.

[Author Response · NeurIPS 2019]

We would like to thank all the reviewers for their comments and feedback. We reply to the reviews in order.

*Move algorithm pseudocode to main text.* We agree it deserves to be part of the main text; we will move it there.

**R#1.** *Theoretical analysis.* Providing theoretical guarantees with function approximation is hard, and few papers do
so in RL. When data goes to infinity and the true value function falls in our family of models, the MC intervals will
converge to the true values and Adaptive TD will be forced to converge too. However, these are asymptotic results.

*Parameter tuning justification.* Adaptive TD seems to be robust to the choice of hyper-parameters. We illustrate this
by simply choosing typical values for the hyper parameters, without doing an exhaustive search for the best values.
For example, we take alpha to be 95% confidence intervals, as this is standard in most statistical work. Even with this
off-the-shelf choice for all environments, Adaptive TD shows strong experimental performance and robustness.

*Performance of different choices of confidence interval $L_{MC}$ and $U_{MC}$.* We can definitely assume distributions **F** other
than Gaussian, or bootstrap estimates, leading to different confidence intervals. We also tried some naive ideas like
setting $L_{MC} = \min_i v_i$ and $U_{MC} = \max_i v_i$. However, this did not perform as well as the Gaussian intervals.

**R#2.** *Add other benchmark algorithms from the field of approximate dynamic programming.* In addition to MC and TD,
in our experiments we also show the most relevant competitor: TD($\lambda$) baselines for a number of different $\lambda$'s.

**R#3.** *The computational complexity increases with time.* We are not sure what is meant here. Our algorithm fits m+1
networks rather than 1, on the same data ($m$ for MC, 1 for TD). Therefore, it requires $m$ times more work. For small $m$
like m=3 or m=5, this is still reasonable. Fortunately, the training of the $m$ MC networks is completely parallelizable.
One can also train the MC networks for much fewer steps (as they tend to converge way faster as they don't bootstrap).

*What is the practical problem that is solved here?* On-Policy evaluation from log data is a very important practical
problem by itself (e.g. recommender systems). Using Adaptive TD within the full RL loop (policy improvement) is left
for future research, as it involves different trade-offs between computation, accuracy and sample efficiency.

*If the MC target is not reliable enough to be used why would these networks' inferred estimates be more reliable?*
Assume first that the function approximation family is rich enough. The MC estimates should be unbiased or, at least,
should have low bias. As a consequence, if the MC target is not "reliable", this means the true underlying variance of
the distribution of values **F**, $\sigma^2$ above (5), must be large. The point is that our goal is not to get accurate MC estimates,
but just to detect when TD estimates are not reasonable. When variance is large, while our MC estimates may not be
reliable, we expect to end up with wide confidence intervals, thus increasing the chances of accepting the TD estimate
as a plausible one. On the other hand, if the MC estimates are biased –so they aren't reliable while the variance can still
be small–, there is no reason to believe MC or TD (usually even more biased) will do a better job than Adaptive TD.

*How much is the algorithm affected by the Gaussianity assumption?* It's important to note that each network is trying
to fit the value function: $V(s) = E_\pi[R(s)]$. While the return distribution $R(s)$ may be complex (e.g. multimodal),
**F** models the disagreement in our predictions for its expectation $V(s)$ as a function of iid training runs. We expect
a smoother and better behaved distribution then, and Gaussianity may be a reasonable assumption. If we still think
Gaussian is a strong assumption, then we have two options. First, if we have a better guess for **F** (e.g. heavy tailed),
we can simply use it to compute the confidence intervals. Second, if the distribution is unknown, we can use a
non-parametric bootstrap approach to approximate the predictive confidence interval directly from our samples $v_{1:n}$.

*How are m, $\alpha$ and the confidence threshold chosen? If for example we pick m to be large, this would create very small*
*confidence intervals which would reduce the algorithm to always using MC estimates (since the TD estimate is always*
*outside)?* This is not true. **F** (line 192) is determined by the training data and algorithm, not by $m$. In particular, the
confidence intervals become narrower with more data, as the value of the true parameter $\sigma^2$ ideally decreases (under a
reasonable training algorithm). Under the Gaussian assumption, $\bar{v}$ and $\hat{\sigma}_m^2$ will tend to their true values for large $m$, and
the TD estimate will need to fall in the confidence interval with respect to the true distribution **F**. Thus, for larger $m$ the
algorithm does not reduce to always using MC. However, when the amount of data tends to infinity, and assuming the
true value function belongs to our family of models, Adaptive TD reduces to MC (as desired). Actually, we did some
experiments with $m = 10, 15$, and results improved (it was also more expensive as we didn't parallelize MC training).

*In the experiments on Lab2D - Figure 2, it seems MC ensemble(3) is not done learning.* There is a misunderstanding
of the plot here. The $x$-axis corresponds to the total training data size (in terms of episodes or rollouts), it does not
correspond to training time or steps. MC and TD take similar training time; Adaptive TD requires $m$ times longer.

*In Figure 3, Adaptive TD(3) behaves very poorly on 2 of the 6 maps.* In maps 0 and 4, in the high-data regime, MC
methods outperform Adaptive TD. This could be expected, as with abundant data MC tends to be best. Maps 0 and 4 are
those where the target region is the furthest from any wall (actually, for map 0 there are no walls at all). In these cases,
due to lack of discontinuities and leakage, the issue we are trying to address in this paper is less severe. Still, in both
cases, Adaptive TD does better than TD (except in map 4 with 100 training rollouts, where it does around 5% worse).

[Meta-Review · NeurIPS 2019]

The argumentation defending the proposed approach, and the numerical evaluation of its performance on realistic examples, are convincing. Despite the fact that the reviewers finally agree on the fact that NeurIPS might not be the best venue for this work, because of the quasi-absence of a theoretical part, I recommend to give it a chance it for the quality of the other dimensions of this work. If the paper is finally rejected, I recommend to the authors to follow the suggestions of the reviews, and to either re-submit to a more speciallized conference, or to consider a theoretical analysis (which can be expected to be rather involved).